# Atomic Layer Deposition of HfO_2_ Films Using TDMAH and Water or Ammonia Water

**DOI:** 10.3390/ma16114077

**Published:** 2023-05-30

**Authors:** Sylwia Gieraltowska, Lukasz Wachnicki, Piotr Dluzewski, Bartlomiej S. Witkowski, Marek Godlewski, Elzbieta Guziewicz

**Affiliations:** Institute of Physics, Polish Academy of Sciences, Aleja Lotnikow 32/46, 02-668 Warsaw, Poland

**Keywords:** ALD, high-k dielectric, HfO_2_

## Abstract

Atomic layer deposition of HfO_2_ from TDMAH and water or ammonia water at different temperatures below 400 °C is studied. Growth per cycle (GPC) has been recorded in the range of 1.2–1.6 Å. At low temperatures (≤100 °C), the films grew faster and are structurally more disordered, amorphous and/or polycrystalline with crystal sizes up to 29 nm, compared to the films grown at higher temperatures. At high temperatures of 240 °C, the films are better crystallized with crystal sizes of 38–40 nm but grew slower. GPC, dielectric constant, and crystalline structure are improved by depositing at temperatures above 300 °C. The dielectric constant value and the roughness of the films have been determined for monoclinic HfO_2_, a mixture of orthorhombic and monoclinic, as well as for amorphous HfO_2_. Moreover, the present study shows that the increase in the dielectric constant of the films can be achieved by using ammonia water as an oxygen precursor in the ALD growth. The detailed investigations of the relationship between HfO_2_ properties and growth parameters presented here have not been reported so far, and the possibilities of fine-tuning and controlling the structure and performance of these layers are still being sought.

## 1. Introduction

Hafnium dioxide (HfO_2_) is known for its many required properties and important technological applications. This significant material exhibits the following unique features, particularly, a high-k dielectric constant, about 20, high permittivity [1], relatively large band gap, about 6 eV [2], a high index of refraction, about 1.91 [3], good thermodynamic and chemical stability, low density of the interface state, low leakage current [4], excellent surface passivation performance, a wide range of UV-IR transparency area, high laser damage threshold [5]and hardness and thermal stability [6]. Bulk HfO_2_ crystallizes in the monoclinic structure at ambient conditions. A phase transformation is observed at a very high temperature above 1000 °C. Another increase of temperature to 2200 °C leads to a tetragonal to cubic transition without diffusion. These conversion temperatures can be significantly transformed by doping, mechanical strains, or manipulation of the surface, so the high-temperature phases are performed in hafnium dioxide thin layers at temperatures much below 1000 °C [7]. For these reasons, HfO_2_ thin films are widely used in many electronic and optical areas in both planar and three-dimensional devices [8,9,10]. Due to the fast progress of optical applications such as plasma screens, emitters of light-emitting diodes, lasers of solid states, and many different scintillators, there is a demand for luminescent materials that are efficient and durable [11]. These applications are often based on wide-gap materials, such as HfO_2_ thin films, with a significant thickness of 200–700 nm and are often activated with rare earth ions [11,12,13]. Also needs to be mentioned that the use of the HfO_2_ insulator indicates promising results for a sensitivity of a modern dielectric modulated FET biosensor for the detection of coronavirus (SARS-CoV-2) in terms of spike, envelope, and DNA proteins of the virus [14]. This may be of great importance in a wide range of applications in medicine. In the production of the electronic industry, there is a demand for higher equipment performance. These demands push the progress of technology forward, which requires a thorough understanding of materials’ chemical, mechanical, and physical properties, such as those found in HfO_2_ used in typical electronic devices.

Atomic layer deposition (ALD) is an excellent method for performing homogeneous HfO_2_ thin layers with high-accuracy thickness control required for applications in electronics, optics, and medicine [15]. In fact, the physicochemical characteristics of HfO_2_ films highly depend on the growth parameters of the ALD process [16]. Film deposition temperature and process time parameters basically specify the dielectric constant, smoothness, purity, and structure of the HfO_2_ layer [17]. It is possible to deposit amorphous as well as polycrystalline HfO_2_ layers. The HfO_2_ films grown by ALD have mainly a monoclinic structure but might also contain tetragonal or orthorhombic phases [18,19].

A variety of chemical precursors were used for the ALD growth of HfO_2_. In most cases, the metal precursor was inorganic, chloride or iodide [18]. All these ALD processes require a relatively high growth temperature in the range of 200–750 °C, typically 300 °C, because, according to the literature data, such a growth temperature allows us to obtain a dioxide with preferably stoichiometric chemistry [20,21,22,23]. On the other hand, the use of inorganic precursors in deposition processes usually causes contamination to be incorporated into the film, such as chlorine or iodine, which obviously affects the thin film’s properties. The high growth temperatures (300 °C and above) can avoid this impurity but cause the creation of high crystalline films with greater surface roughness. The etching of the growing oxide surface by the precursors themselves also causes additional surface roughness and nonconformity [24]. The growth of HfO_2_ from inorganic precursors at 300 °C can result in amorphous growth, while at higher temperatures, such as 600 °C, the obtained films are polycrystalline [17,18].

Precursors with high reactivity and completely self-limiting surface reactions are needed for high homogeneity and low-temperature deposition [15]. The metal-organic chemical compounds of hafnium, such as hafnium amides, have thermal stability and variability, making them suitable for use in ALD processes [25,26]. Compared to metal chloride and iodide, the organometallic precursors are much more reactive because the metal-chloride or metal-iodide bond is much stronger than the metal-nitrogen bond (both of which are weaker than the metal-oxygen bond) [24,27]. As described in previous works [28,29,30], conformal, uniform and smooth HfO_2_ films can be deposited using hafnium organic precursors and limit the growth temperature to as well as 100 °C, bearing in mind the performance of these layers on the flexible substrates. It was also shown that, for deposition temperatures above 100 °C, structures of hafnium films are polycrystalline, while for a lower temperature, film structures are amorphous. As mentioned above, similar dependence exhibits ALD layers from metal-inorganic precursors.

The growth process of HfO_2_ films has a strong influence on their properties [17]. Therefore, it is not trivial to predict whether a layer with specific properties will be formed in a given technology, but these need to be experimentally determined. According to our knowledge, the relationship between GPC, crystallinity, morphology, composition, electrical properties and deposition temperature for ALD-grown HfO_2_ thin films has not been studied simultaneously. Data about such a detailed comparison for HfO_2_ obtained from metal-organic precursors over a wide range of ALD temperatures have been published so far. In this work, we report on ALD of hafnium oxide thin layers from (CH_3_)_2_N_4_Hf and water (deionized and/or ammonia water) at temperatures in the range of 85–350 °C onto silicon and glass substrates. The structural and electrical properties of the layers are studied with X-ray diffraction, atomic force microscopy, energy-dispersive X-ray spectroscopy, transmission electron microscope, and electronic measurements.

## 2. Materials and Methods

### 2.1. ALD Preparation of HfO_2_ Films

ALD of highly conformal HfO_2_ films is achieved using metal alkylamide hafnium precursor tetrakis(dimethylamido)hafnium (TDMAH) (Merck KGaA, Darmstadt, Germany). This metal precursor is very reactive, with hydroxylated surfaces covered with water vapor [31]. Deionized water and/or ammonia water as a twenty percent solution of NH_3_ in H_2_O were used as an oxygen source during this experiment. HfO_2_ layers were grown on (100) oriented n-Si and glass substrates using a Savannah-100 reactor from Ultratech/CNT (Veeco, San Jose, CA, USA). An important advantage of the ALD method is the self-limiting and sequential growth process, which allows the use of very reactive precursors and deposition at relatively low temperatures in the range of 85–350 °C. In this temperature range, all oxides were obtained with deionized or ammonia water as an oxygen precursor. It has been found that nitrogen is beneficial to the oxide with a high dielectric constant [32]; therefore, a part of the HfO_2_ layers was grown at 200 °C by replacing deionized water with ammonia water in each ALD cycle (defined as “100% H_2_O:NH_3_”) and in every second ALD cycle (defined as “50% H_2_O:NH_3_”). Hafnium oxide layers were created as a product of the double-exchange chemical reactions numbers 1 and 2.

With deionized water as:Hf[(CH_3_)_2_N]_4_ + 2H_2_O → HfO_2_ + 4HN(CH_3_)_2_,(1)
and with ammonia water as:Hf[(CH_3_)_2_N]_4_ + NH_3_・2H_2_O → HfO_2_ + 4HN(CH_3_)_2_ + NH_3_,(2)

According to these reactions, as described in our previous paper [33], in the ALD process, two precursors were led into the growth chamber of the reactor alternately using the continuous flow of N_2_ gas as a carrier. After each precursor pulse, the reactor was purged with pure N_2_ gas. The growth parameters, such as pulse times of the precursors and purging times after precursor pulses, were the same for all depositions of selected layers. The optimum pulse times of the precursors for self-saturated and homogeneous deposition were very short, in the range of 40–100 milliseconds. These films were grown using sequential pulsing of the first precursor and second precursor, with purging periods of about 10 s for both precursors. Thus, the duration of a complete ALD cycle was about 20 s.

Besides deposition temperature and the precursors, film properties can be modified by pulsing and purging times. It has been found that at relatively low growth temperatures, the crystal structure of oxide layers can be changed by the reduction of the purge time between the precursor pulses [16]. Therefore, for layers grown at 200 and 350 °C, the lengths of purge periods between precursor pulses with pure N_2_ gas were shortened to about 5 s, which is defined as a “short purge.”

### 2.2. Materials Characterization 

The so-obtained oxide layers were characterized by a range of experimental methods with measurement details similar to our previous papers [29,33]. The structure and crystallographic orientation of the layers obtained on glass substrates were determined by X-ray diffraction (XRD) using a system equipped with an X-ray mirror and a two-bounce monochromator at the incident beam. The diffracted beam was measured over an angular range of 2θ from 20 to 65 degrees with a 2-dimensional solid-state X-ray detector, PIXcel (Malvern Panalytical Ltd., Malvern, United Kingdom). Crystal sizes of HfO_2_ films were determined by the XRD data for the line broadening of the peak (2θ = 34°) using the Scherrer equation [34]. The structure of hafnium layers was also investigated by transmission electron microscopy (TEM) with the use of an FEI Titan 80–300 Cubed microscope (FEI, Lausanne, Switzerland) at a 300 kV accelerating voltage. The surface and cross-section images with the thicknesses of the films were obtained using a scanning electron microscope (SEM; Hitachi SU-70, Tokyo, Japan) at an operation voltage of 15 kV. Chemical analysis was performed using an energy-dispersive X-ray spectrometer (EDX) at 5 kV accelerating voltage, a piece of additional equipment for electron microscopes. The EDX analysis was carried out using a Thermo Scientific UltraDry silicon drift X-ray detector of SEM (Waltham, MA, USA) and a Noran 7 X-Ray Microanalysis System. The surface morphology was examined by atomic force microscopy (AFM; Bruker Dimension Icon, Billerica, MA, USA) using the PeakForce Tapping mode and silicon nitride probes with sharp tips (a tip radius: 2 nm). Surface roughness and particle size were determined by a root-mean-square (RMS) roughness and an average particle height on the surface of HfO_2_ layers, respectively. These parameters were obtained using AFM height measurements from images taken from a 10 × 10 μm^2^ region. 

Basic electro-physical properties such as current-voltage (I–V) and capacitance-voltage (C–V) measurements and dielectric constant values of HfO_2_ layers were determined for MOS-type capacitors. The 100 nm thick HfO_2_ films grown at 85 °C were used as a gate dielectric for MOS structure fabrication for C–V measurements, whereas dielectric constant values were estimated for MOS structures with 100–740 nm thick HfO_2_ films deposited at temperatures in the range of 85–350 °C. The duration of a complete ALD process for these layers was between 4 and 24 h, depending on film thickness. Gate dielectrics in MOS structures were obtained on n-Si substrates (10^−2^ Ωcm resistivity). Titanium/gold was used as metal gate electrodes with areas of 0.04–0.09 cm^2^. These gate electrodes were deposited by sputtering. Relative dielectric constant values were calculated from capacitance measurements varying with applied bias voltage at a frequency of 1 MHz. The capacitance value measured at accumulation was used to extract the dielectric constant of HfO_2_ ($k_{HfO_{2}}$, also referred to as relative permittivity). 

For these conditions, the MOS structure can be considered as a parallel-plate capacitor (without quantum mechanical effects from a substrate and a gate) [35]. The energy band diagrams for the metal/HfO_2_/n-Si system are schematically shown in Figure 1a,b for gate biasing at accumulation and depletion, respectively. When a MOS structure is biased with positive or negative voltages, specific cases may exist at the semiconductor surface and the interface between the semiconductor and insulating oxide [36]. When the capacitor is biased with a positive voltage (V_G_ > 0 V), the majority of carriers (electrons for the n-type Si substrate) tunnel from n-Si and are accumulated near the semiconductor surface at the interface between the semiconductor and insulating oxide, as shown in the schematic energy band diagram in Figure 1a. Furthermore, when the voltage is biased with a small negative voltage (V_G_ < 0 V), the electrons near the semiconductor surface at the interface are depleted, as shown in Figure 1b. These relations are then used to derive the C–V characteristic for the HfO_2_ dielectric layer in the MOS structure [36].

Knowing the thickness of the layer, the dielectric constant of the HfO_2_ layer can be calculated using the following equations:(3)$$k_{HfO_{2}}=\frac{C_{HfO_{2}}t_{HfO_{2}}}{A_{G}\varepsilon_{0}}$$
then the HfO_2_ dielectric capacitance density ($C_{HfO_{2}}$ in F/cm^2^) is expressed as:(4)$$C_{HfO_{2}}=\frac{C}{A_{G}}$$
where *C* is the accumulation capacitance of the test structure, $A_{G}$ is the area of the capacitor, $t_{HfO_{2}}$ is the HfO_2_ layer thickness, and $\varepsilon_{0}$ is the electric permittivity of vacuum (=8.84 × 10^−14^ F/cm) [35]. The flat band capacitance is: (5)$$C_{FB}=A_{G}\frac{C_{HfO_{2}}C_{sFB}}{C_{HfO_{2}}+C_{sFB}}$$
then the flat band capacitance of Si is:(6)$$C_{sFB}=\frac{\varepsilon_{0}k_{s}}{L_{D}}$$
where $k_{s}$ is the dielectric constant of Si, $L_{D}$ is the Debye length $\sqrt{\frac{kT}{q}\times \frac{\varepsilon_{0}k_{s}}{qN}}$, k is Boltzmann’s constant (8.61 × 10^−5^ eV/K), T is the temperature (300 K), q is the electron charge (1.60 × 10^−19^ C), and N is the donor concentration in Si (10^18^ cm^−3^) [36]. For ideal MOS structures, at $V_{G}=V_{FB}=0$ applied voltage on the metal gate (where $V_{FB}$ is the flat band voltage, i.e., the voltage required to bring the Fermi levels into alignment), the work function difference between the metal and n-type semiconductor (Si), $\Phi_{MS}$, is zero, therefore:(7)$$\Phi_{MS}=\Phi_{M}-\left(\chi +\frac{E_{g}}{2q}-\Psi_{B}\right)=0$$
where $\Phi_{M}$ is the metal work function, χ is the Si electron affinity, $E_{g}$ is the Si band gap, and $\Psi_{B}$ is the potential difference between the Fermi level E_F_ and the intrinsic Fermi level, E_I_ [36]. In a real MOS structure:(8)$$V_{FB}=V_{G}=\Phi_{MS}\pm \frac{Q_{f}}{C_{\;}}$$

Thus, a value for fixed charge density $Q_{f}$ can be determined from measured values of C and from calculated $V_{FB}$ and $C_{FB}$. The sign of the fixed charge is also important, as the negative fixed charge correlated with the plus sign in Equation (8), and the positive fixed charge correlated with the minus sign [35].

Electrical measurements were carried out by Keithley 2601 A (Tektronix, Inc., Beaverton, OR, USA), 236 electrometers and 4275A Multi-Frequency LCR Hewlett-Packard (Hewlett-Packard, Palo Alto, CA, USA). The structural (without XRD measurements), morphology, chemical analysis, thickness and electrical properties of layers were measured for the films grown on silicon substrates. All presented ALD processes were optimized to obtain highly uniform and conformal hafnium films with a thickness of around 100 nm in the case of TEM and C–V measurements and with a thickness in the range of 520–740 nm in the case of other measurements. All measurements were performed at room temperature and on the films in their as-grown state.

## 3. Results and Discussion

### 3.1. Film Thickness and Growth per Cycle

Figure 2 presents the dependence of the increase of film thickness per cycle, referred to here and often in ALD literature as GPC, on the growth temperature for HfO_2_. It shows that the GPC value for layers deposited from TDMAH and H_2_O decreases from around 1.4–1.2 Å per cycle when the temperature rises from 85 to 240 °C. This downward trend can be explained by the decrease in the concentration of -OH groups participating in the surface reaction on the HfO_2_ surface [16]. The GPC value stabilizes at 1.3 and 1.2 Å per cycle when the growth temperature is in the range of 100–135 °C and 180–240 °C, respectively. The increase of GPC below 100 °C is possibly due to the precursor condensation [37]. However, when the growth temperature is 350 °C, the GPC increases to 1.3 Å per cycle, possibly because the ligands of the precursor are decomposed by high thermal energy, resulting in more effective chemical reactions [38]. In other words, at deposition temperatures over 240 °C, high thermal energy decomposes the ligand, which increases the GPC [38]. The results show that the growth does not depend on the type of oxygen precursor used (H_2_O or H_2_O:NH_3_) since, in both cases, the GPCs are almost the same, 1.2 Å/cycle at 200 °C. Alternatively, for the growth of HfO_2_ layers with purge times shortened from 10 to 5 s (as described previously) at temperatures of 200 and 350 °C, the GPC increases from 1.2 to 1.3 and from 1.3 to 1.6 Å/cycle, respectively. It can be considered that the ALD process with very short purge times causes some reagents to remain in the chamber, resulting in a reaction similar to the CVD, which increases the GPC [38]. The thickness of the films from TDMAH and H_2_O is changed from around 650 to 520 nm with the increase of temperature from 85 to 240 °C. When the deposition temperature is enhanced to 350 °C, the film thickness achieves approximately 600 nm, while the thickness of the films from TDMAH and H_2_O:NH_3_ grown at 200 °C is approximately 550 nm. The growth of HfO_2_ layers at temperatures of 200 and 350 °C with short purge times results in the increase of the thickness to 590 nm and to 740 nm, respectively. In addition, any significant changes in the GPC value are not observed for thicker HfO_2_ films (with more ALD cycles). 

### 3.2. Chemical Component

EDX measurements were performed to calculate the chemical composition of above 500 nm-thick HfO_2_ layers obtained at temperatures in the range of 85–350 °C, confirming that the ALD layers contained Hf and O as well as C and N elements (Table 1). A main source of carbon and nitrogen is the deposition process, which involves water (H_2_O and H_2_O:NH_3_) as an oxidant, metal-organic precursor ((CH_3_)_2_N_4_Hf) and N_2_ as a carrier gas that is able to leave the rest of the carbon and nitrogen residues in the HfO_2_ layers. These impurities can create compounds with oxygen (e.g., CO, CO_2_, NO, NO_2_, etc.) and/or with hydrogen (e.g., CH and NH). The carbon content in the obtained films is about 6%. Therefore, we cannot eliminate that carbon contamination generally comes from the surface of the films because of the specification of EDX measurements. The nitrogen content in the examined layers is about 4.5%. It shows that the N content only slightly increases when the ALD processes are carried out using ammonia water. Furthermore, an increase in light element contents for HfO_2_ layers grown below 100 and above 240 °C could be connected to the higher GPC and possibly to the precursor concentration and the thermal precursor decomposition, respectively [38]. However, the use of short purge times of nitrogen gas led to higher GPC and lower carbon and nitrogen content but also oxygen reduction in the film. These aberrations between the measurement data and the general tendency are probably due to kinetic reasons, i.e., incomplete reactions due to the short nitrogen purge times [39,40]. At the same time, the oxygen to metal (O:Hf) atomic ratio with a value ranging from 1.94 to 2.03 in other analyzed films (except layers with short purge times) is close to the desired stoichiometry of HfO_2_ films. The calculated O:Hf ratio of 2.03 shows the excess of oxygen atoms at growth temperatures lower than or equal to 100 °C, most probably due to significant hydroxylation of the surface. The concentration of O to Hf in the HfO_2_ layers deposited at higher temperatures between 100 and 240 °C is similar and equal to 1.94. The decrease in the O:Hf ratio may result in an increase in the concentration of oxygen vacancies in the HfO_2_ [41,42]. In the previous papers in Refs. [33,43], it was also noted that oxygen content decreased with the increasing deposition temperature. However, the layers grown at 350 °C demonstrate an O:Hf ratio equal to 2. This stoichiometric dioxide can correspond to the high-quality crystalline structure of the examined hafnium films.

### 3.3. Crystal Structure and Surface Morphology

The X-ray diffraction results of 500 nm thick HfO_2_ layers deposited at temperatures between 85 and 350 °C with TDMAH-H_2_O are shown in Figure 3, while the particle sizes on the surface and crystal sizes determined from AFM and XRD measurements of these HfO_2_ films are presented in Table 2. The large background in the diffraction patterns is due to the apparatus settings or the substrates. The XRD curve of the HfO_2_ layer deposited below 100 °C does not exhibit any sharp diffraction peaks, typical for crystal structure, which suggests that the obtained layer is in an amorphous and/or poorly crystallized form [9], as stated in our previous works [29,30]. When the growth temperature increases to 100 °C, a few small peaks appear in the XRD curve of the HfO_2_ layer, pointing out that the film is being transformed into a polycrystalline state with a predominantly monoclinic phase, with a small orthorhombic phase contribution, but is probably still generally in an amorphous form, which can be recognized to the presence of small crystals of about 29 nm in an amorphous matrix [9]. In the temperature range of 135–240 °C, the XRD pattern of the films demonstrates characteristic peaks of the monoclinic and orthorhombic phases, signifying the transformation of layers to the polycrystalline state, which corresponds to significantly larger crystal sizes of 38–40 nm. The crystallites are larger at higher growth temperatures, which suggests improved density and crystallinity of HfO_2_ films [9,44]. Moreover, when the growth temperature increases to 350 °C, the XRD pattern shows sharp diffraction peaks characteristic for only one crystal phase of monoclinic HfO_2_ with a marginally smaller crystal size of 39 nm. This slightly smaller crystal size of the monoclinic HfO_2_ films, compared to the mixture of monoclinic and orthorhombic HfO_2_ films grown at 240 °C, can be clarified by the enhanced concentration of carbon and nitrogen, which induces voids and hydrogen-related defects in the films [45]. To form a crystalline phase, the deposited atoms should have sufficient energy that allows the atom to move into a low-energy position, which leads to the creation of a crystalline phase. High growth temperature is able to produce energy to perform crystalline phases [9].

The relevant XRD results concerning crystal structures of hafnium films with thicknesses of 100 nm and above 500 nm are confirmed by TEM measurements presented in Figure 4 and Figure 5 and cross-section SEM images in Figure 6. TEM results were performed for 100 nm thick HfO_2_ layers deposited at three temperatures of 85, 135, and 350 °C. SEM images were measured for HfO_2_ layers thicker than 500 nm at selected temperatures in the range of 85–350 °C. As noted in our previous paper [29] and as shown in Figure 4 and Figure 6, when the growth temperature exceeds 100 °C, the HfO_2_ films are polycrystalline and reveal granular structures. Thus, the HfO_2_ films, obtained at lower temperatures, are amorphous. TEM and SEM images also indicate that HfO_2_ obtained at high growth temperatures (350 °C) are of better crystalline quality corresponding to a preferentially oriented columnar structure, in contrast to films grown at lower temperatures. At higher growth temperatures, the columnar structure increases its density.

In addition, Figure 5 shows that amorphous HfO_2_ layers grown at low temperatures (<100 °C) reveal a well-defined, smooth dielectric/semiconductor HfO_2_/SiO_2_/Si interface. The thickness of this interlayer does not exceed 1 nm. The presence of a very thin amorphous SiO_2_ layer results from the oxidation of the silicon substrate due to contact with air. On the other hand, the growth temperatures higher or equal to 100 °C may induce a reaction between the crystalline HfO_2_ film and the semiconductor substrate; therefore, the formation of an interfacial layer takes place during the deposition process. This amorphous interlayer with a thickness of a few nanometers (3–5 nm) contains hafnium, oxygen and silicon.

XRD and TEM characterizations revealed that the HfO_2_ layers deposited at temperatures below 100 °C are amorphous, whereas layers grown at temperatures higher/equal to 100 °C are polycrystalline. The preferred orientation of the layer is various at different temperatures; see Figure 3, Figure 7 and Figure 8. The XRD curves show that all layers mostly crystallize in monoclinic phases [44]. For the films grown at 100–240 °C, the (271), (020) and (221) diffraction peaks were recorded in addition to the reflections of the monoclinic phase, but with much lower intensities than the reflections from the monoclinic phases. These former reflections may be assigned to the orthorhombic HfO_2_ [44,46]. The monoclinic crystallization of the HfO_2_ layers was expected based on the phase-stability considerations [47]. The (111) and (−111) peaks should be the most intense in comparison to the powder diffraction database of monoclinic HfO_2_ [48], but the recorded curve was not observed [44]. The (020) and (200) reflections for the monoclinic phase and (020) and (221) for the orthorhombic phase are the most intensive, while the comparative intensity of these reflections depends on the deposited temperature [44]. For the monoclinic phase, the (020) peak has the strongest intensity in the films grown at 350 °C, while the (200) reflection is the highest in the films grown at 350 and 240 °C. In turn, for the orthorhombic phase, the (020) reflection shows the highest intensity in the films deposited at 240 °C, while the (221) peak is the strongest in the films deposited at 200 °C. At the 100–135 °C range, mainly the (020), (021) and (130) diffraction peaks from the monoclinic phase and also other small (020) peaks from the orthorhombic phase emerge first. However, in the 180–200 °C range, the relative intensities of reflection belonging to both monoclinic and orthorhombic phases can be clearly seen. The (020), (200), (021), (130), (020), and (221) diffraction peaks of monoclinic and orthorhombic HfO_2_ phase, respectively, have a significant influence on the crystal structure. At 240 °C, the dominant phase became monoclinic, as confirmed by similar and relatively intensive (020) and (200) reflections. Mainly one position of the (020) reflection corresponding to the orthorhombic HfO_2_ phase is recorded. Worth noticing that a combination of monoclinic and orthorhombic phases is observed in HfO_2_ films obtained at 100–240 °C. These properties, related to the changes in crystal structure with increasing growth temperature, have been explained by the increase in crystallinity and density of monoclinic HfO_2_ in films. This confirms the increase of the crystallite size and preferred orientation of the film along (020) with increasing growth temperature [49]. Therefore, at 350 °C, HfO_2_ films show the focus along the (020) direction structure with only one structural, monoclinic phase. This indicates that the crystallization temperature of monoclinic HfO_2_ is above 300 °C. These HfO_2_ films show a strong (020) orientation, which may result from the lowest surface free energy, but there are some other smaller peaks, which may be a consequence of the preferential orientation of other crystal directions [9]. Consequently, the increase in growth temperatures favored the preferred orientation along (020) whereas reducing the strain energy in the HfO_2_ films [9,50]. Orthorhombic HfO_2_ has been observed below 300 °C. It is possible that this phase is stable in small crystallites at lower temperatures and is transformed into the monoclinic phase with the increase of crystallite sizes [44]. Possibly, as the film grows, orthorhombic nanocrystallites form at first, but as the growth goes on, the monoclinic phase becomes dominant [51].

The low-temperature ALD (below 100 °C) causes the formation of amorphous HfO_2_ films with very small particles on the surface (2 nm) and a flat surface (RMS = 1 nm), see Table 2 and Figure 9. When the growth temperatures are higher in the 100–350 °C range, the HfO_2_ layers have been transformed into polycrystalline states with more oriented structures. In general, as the deposition temperature rises from 100 to 350 °C, the particle sizes on the surface and RMS surface roughness decrease from 83 to 17 nm and from 21 to 4 nm, respectively. Thus, this significant correction of the surface smoothness can be explained by the rearrangement of atoms resulting from higher growth temperatures, leading to the improved structure and packing density, as well as the reduction of the roughness of the films [52]. These results are also consistent with the transition from amorphous through polycrystalline HfO_2_ with two structural phases and further to polycrystalline HfO_2_ with one phase at higher deposition temperature, as shown by XRD patterns and TEM measurements. 

Furthermore, the use of short nitrogen purge times has a significant effect on the dominant orientation of the monoclinic phase because it leads to decreasing reflection intensity and the presence of peaks from monoclinic and orthorhombic phases in the XRD pattern, as presented in Figure 7 and Figure 8. This is due to the fact that the use of short purge times reduces structural order and determines a more randomly oriented polycrystalline structure of HfO_2_ films which corresponds to the change in grain sizes, the increase in RMS surface roughness and particle sizes on the surface (see Figure 10 and Figure 11 and Table 2). As already noted, these changes are most probably due to the reactions in the ALD processes that may not be thoroughly completed with short nitrogen purge times. 

The XRD patterns, calculated nanocrystallite sizes and cross-section SEM images in Figure 7 and Figure 12 and Table 2 for HfO_2_ films deposited with ammonia water are similar to the structural properties of HfO_2_ films deposited with deionized water. Additionally, Figure 13 proves that HfO_2_ films grown with ammonia water are characterized by an RMS surface roughness of 10 nm, like HfO_2_ films deposited with deionized water. However, HfO_2_ films with identical RMS values could have different surface morphology. The reason is that statistical measurements, like RMS surface roughness, are restricted to vertical, not horizontal, structures [53]. In this case, the use of H_2_O:NH_3_ as an oxygen precursor in ALD processes for hafnium layers results in the increase in particle size on the HfO_2_ surface from 33 to 38 nm and the decrease in crystal size from 39 to 37 nm, which is probably a consequence of inhibited crystallization and defects related to nitrogen by the introduction of the -NH_x_ groups from ammonia water to these oxide films [32,45]. It can be assumed that, as a result of ALD reactions with H_2_O:NH_3_ as an oxygen source, the -NH group is introduced exactly at the oxygen site of the HfO_2_ structure, as similarly presented in our previous work [54].

The above results demonstrate that the growth temperature and the length of the nitrogen purge times play a main role in the growth of thin films [9].

### 3.4. Dielectric Properties 

After establishing the relationship between selected growth parameters, crystal structure, roughness of surface and crystal size, this part focuses on the dielectric properties of HfO_2_ films. To confirm the influence of the crystal phase on the dielectric constant., the $k_{HfO_{2}}$ values calculations are extracted from the C–V curves for HfO_2_ films thicker than 500 nm and grown at various temperatures between 85 and 350 °C. The results of these calculations are shown in Figure 14. The measured dielectric constants fall within a range of 19.8–30.5. For amorphous HfO_2_ layers deposited at 85 °C, the relatively high dielectric constant is below 20, which visibly relates to the nonpresent or restricted interfacial film between a substrate and the growing HfO_2_ layer [50], as well as grain boundaries or surface facets in the HfO_2_ film. These $k_{HfO_{2}}$ values are comparable to parameters described in the papers for crystalline HfO_2_ materials, which leads to the conclusion that ALD processes ensure the growth of dense dielectric layers with a low pinhole density [1]. When the growth temperature increases, the HfO_2_ films become more crystalline, and the dielectric constant rises even above 30 for 350 °C. In this regard, the increase of dielectric constant values confirms the improvement of crystallinity and crystal size and, in turn, the increased dipole density [55,56]. Moreover, the decrease in the $k_{HfO_{2}}$ values of the films deposited with short nitrogen purge times confirm the structurally more disordered films compared to HfO_2_ obtained with long purge times. The effect of using ammonia water as an oxygen precursor in the ALD growth on the dielectric constant values of HfO_2_ is investigated as well. It is clearly demonstrated that in this case, the dielectric constant of hafnium oxide is enhanced, perhaps because nitrogen in the form –NH_x_ groups reduces the diffusion of impurities inside the oxide layer [32].

The structural properties of HfO_2_ described above are correlated with the electrical parameters of MOS capacitors with insulators in the form of polycrystalline layers (deposited above 100 °C) and amorphous layers (deposited below 100 °C). Furthermore, the polycrystalline structure formed at higher growth temperatures, as noticed in our previous paper [29], may provide leakage paths in dielectric films. This is evident in the I–V characteristics, where the leakage current is enhanced in polycrystalline films compared to amorphous ones [50]. The properties for I–V characteristics of MOS structures with the 100 nm thick amorphous dielectric oxides are described in detail in our previous paper, Refs. [28,29,30]. The growth temperature of these layers was 85 °C. The HfO_2_ dielectric layers analyzed in this work exhibit good insulating properties, a dielectric strength of ~1 MV/cm (as compared to the 5/90/5 nm thick composite layer of Al_2_O_3_/HfO_2_/Al_2_O_3_~4 MV/cm), and a leakage current of ~2 × 10^−6^ A/cm^2^ at 1 V (as compared to composite layer in the range of 10^−8–^10^−9^ A/cm^2^). 

These MOS structures were characterized by high-frequency C–V measurements. The C–V curves are stable, which refers to the homogeneous nature of these ALD films grown at low temperatures. The normalized C–V characteristics of HfO_2_ and the composite HfO_2_ with Al_2_O_3_ layer are presented in Figure 15a,b, respectively. The curve measured for HfO_2_ shows a typical C–V characteristic with an accumulation region for the moderate positive gate voltage and a depletion and also an inversion region for the negative gate voltage (Figure 1). The MOS capacitor with HfO_2_ is biased at accumulation for 1V and at depletion for −1.5 V. The largest value of measured capacitance density in accumulation is $C_{HfO_{2}}$~185 nF/cm^2^, which corresponds to the dielectric constant of $k_{HfO_{2}}$~21 ± 3 (as compared to composite layers ~160nF/cm^2^ and ~19 ± 3, respectively). From the measured C–V data, the calculated flat band capacitance value ($C_{FB}$~1.8 nF) and the flat band voltage value ($V_{FB}$~−1.4 V), a positive fixed charge density of $Q_{f}$~1.5 × 10^12^ cm^−2^ is obtained. In comparison to the mentioned HfO_2_ properties, the C–V characteristic of the composite HfO_2_ with Al_2_O_3_ layer is different, as shown in Figure 15b, at the depletion region for positive gate voltage. Consequently, from this measured data and the calculated $C_{FB}$ value (~1.5 nF) and $V_{FB}$ value (~1.9 V), a negative fixed charge density of $Q_{f}$~2 × 10^12^ cm^−2^ is obtained. From this analysis, the HfO_2_ layers have higher dielectric constants and lower fixed charge densities but worse insulating properties than the composite HfO_2_ with Al_2_O_3_ layers, which may be caused by a high density of interface states and charges [57]. The ALD-HfO_2_ exhibits good electrical properties and a relatively large dielectric constant, as required for practical applications [58].

## 4. Conclusions

In summary, uniform, tight and stable HfO_2_ films can be successfully grown by reactive ALD at temperatures ranging from 85 to 350 °C using TDMAH as Hf precursor and deionized and/or ammonia water as O precursor. This metal precursor is confirmed to be very reactive with both deionized and ammonia water, provided a saturating dose of the metal precursor is used. These results show that the deposition temperature and length of the nitrogen purge time play an important role in the ALD growth of thin dielectric films. The HfO_2_ films are found to contain different crystalline structures. Layers grown at deposition temperatures lower than 100 °C are amorphous, while thicker layers fabricated at temperatures higher or equal to 100 °C are polycrystalline with one or two crystalline phases. A substantial reduction of surface roughening occurs both for crystalline layers grown above 300 °C and for amorphous layers grown below 100 °C. It is worth noticing that the improved crystallinity leads to increased dipole density, which results in enhanced relative permittivity (the dielectric constant value). These films are more densely packed. The change of the HfO_2_ films obviously affects the electrical properties of layers because the phase affects the strain and surface roughness of films as well as the dielectric constant.

These above studies also indicate that the electrical properties of the HfO_2_ layers can be effectively tailored by the microstructure of the films, which strongly depends on the ALD processing conditions. The change of ALD parameters during the growth results in the modification of structural and electrical properties of the studied HfO_2_ layers. As a result, amorphous to crystalline phase HfO_2_ thin films can be obtained with relatively high dielectric constants.

## Figures and Tables

**Figure 1 materials-16-04077-f001:**
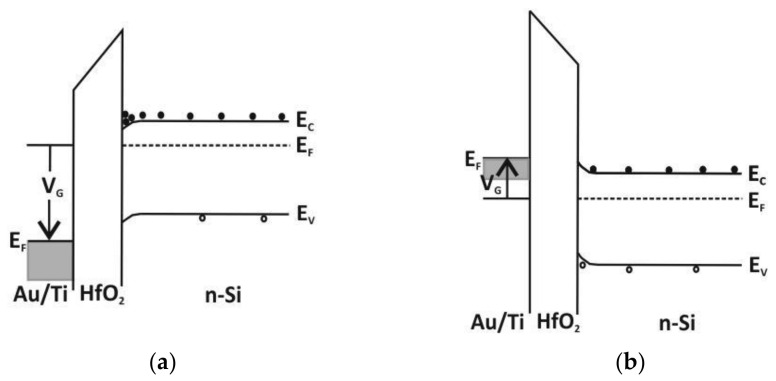
The energy band diagrams of hafnium dielectrics for the metal/HfO_2_/Si system for (**a**) accumulation for a moderate positive gate voltage of V_G_ > 0 V and (**b**) depletion for a moderate negative gate voltage of V_G_ < 0 V, where E_F_ is the Fermi level, E_v_ is the valence-band edge, E_c_ is the conduction-band edge, V_G_ the applied gate voltage.

**Figure 2 materials-16-04077-f002:**
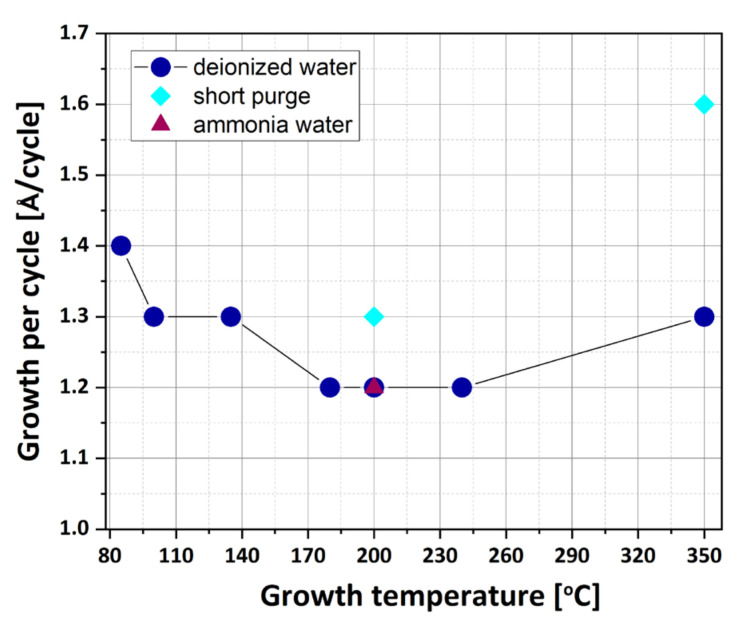
Growth per cycle (GPC) of HfO_2_ films as a function of growth temperature.

**Figure 3 materials-16-04077-f003:**
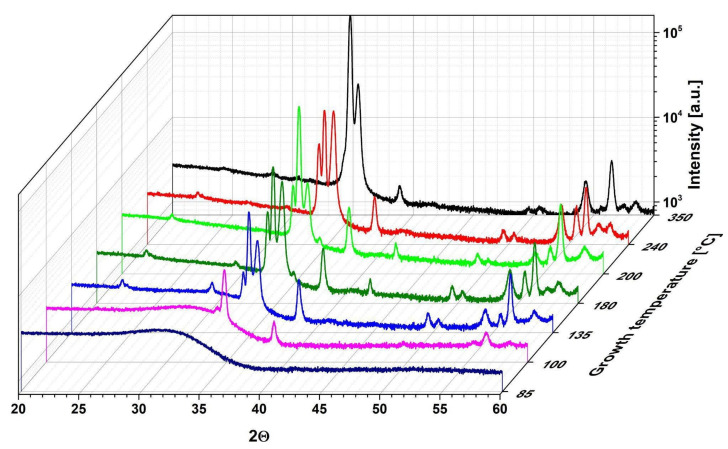
XRD patterns of above 500 nm-thick HfO_2_ films on glass substrates deposited at different growth temperatures in the range of 85–350 °C from TDMAH and H_2_O.

**Figure 4 materials-16-04077-f004:**
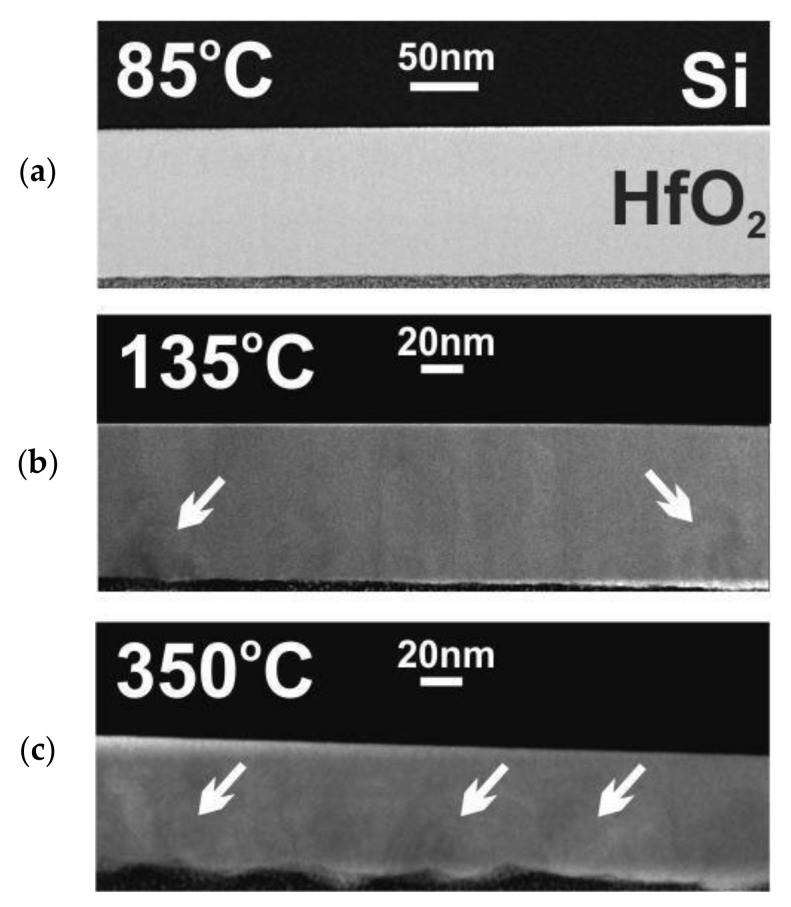
TEM images of HfO_2_ films with thicknesses of about 100 nm. The regions with the HfO_2_ crystal structures indicated by white arrows. Layers were grown at substrate temperatures of (**a**) 85, (**b**) 135 and (**c**) 350 °C on a Si substrate.

**Figure 5 materials-16-04077-f005:**
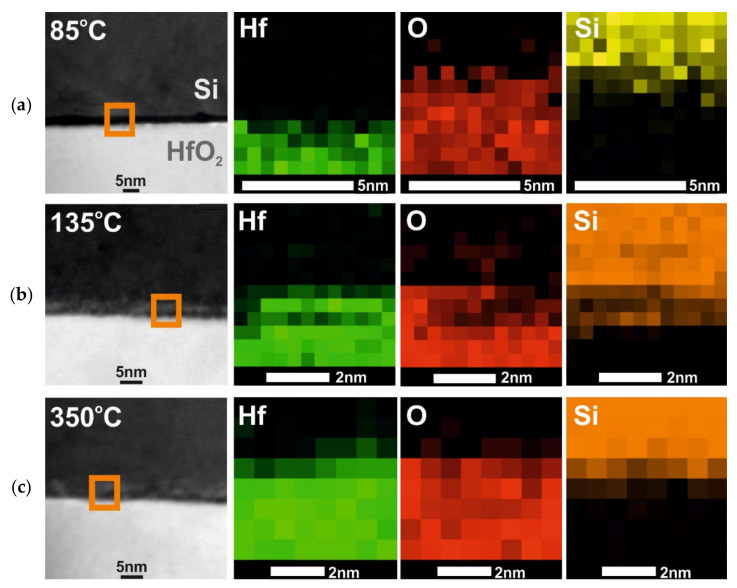
TEM images (**left**) and element distribution maps of the orange square areas obtained by EDX of Hf (green color), O (red color) and Si (orange color) of HfO_2_ films with a thickness of about 100 nm. The orange square areas show dielectric/semiconductor interfaces. Layers were grown at substrate temperatures of (**a**) 85, (**b**) 135, and (**c**) 350 °C on Si substrates.

**Figure 6 materials-16-04077-f006:**
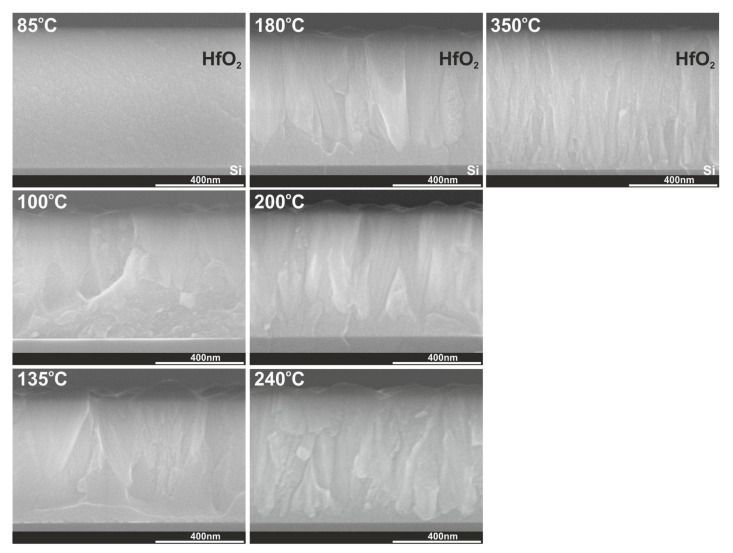
Cross-section SEM images for HfO_2_ thin films with thicknesses above 500 nm. Films were grown on a Si substrate at a temperature in the range of 85–350 °C.

**Figure 7 materials-16-04077-f007:**
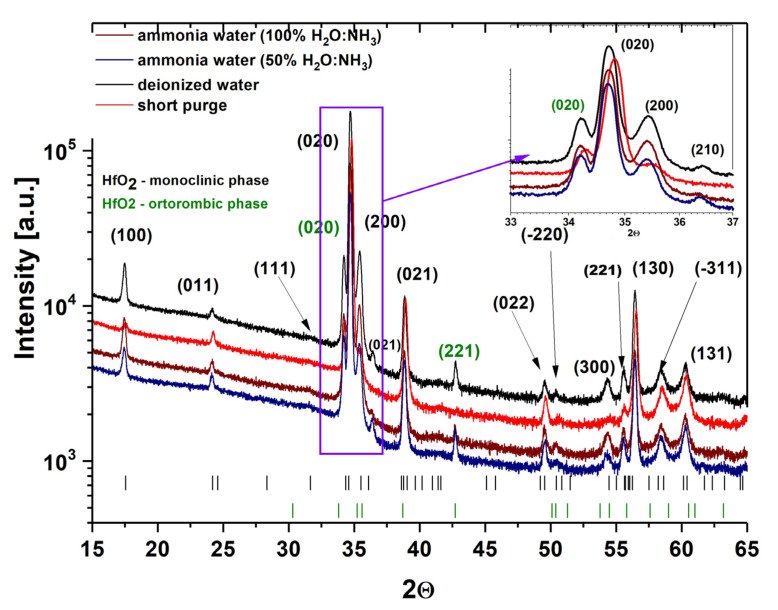
XRD patterns of HfO_2_ thin films with thicknesses above 500 nm deposited at 200 °C on a glass substrate. The peak positions from crystalline phases are shown at the bottom.

**Figure 8 materials-16-04077-f008:**
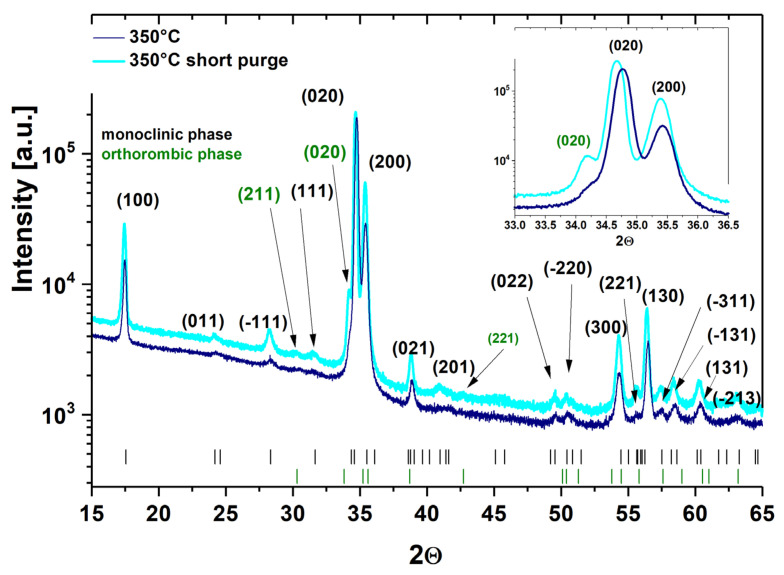
XRD patterns of HfO_2_ thin films with thicknesses above 500 nm deposited at 350 °C on a glass substrate. The peak positions from crystalline phases are shown at the bottom.

**Figure 9 materials-16-04077-f009:**
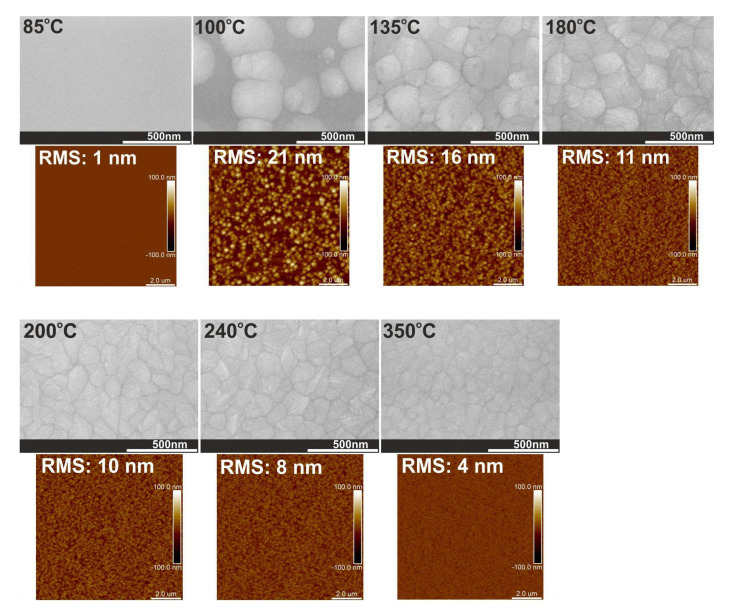
SEM images and AFM surface morphology with RMS values for HfO_2_ thin films with thicknesses above 500 nm. Films were grown on a Si substrate at a temperature in the range of 85–350 °C.

**Figure 10 materials-16-04077-f010:**
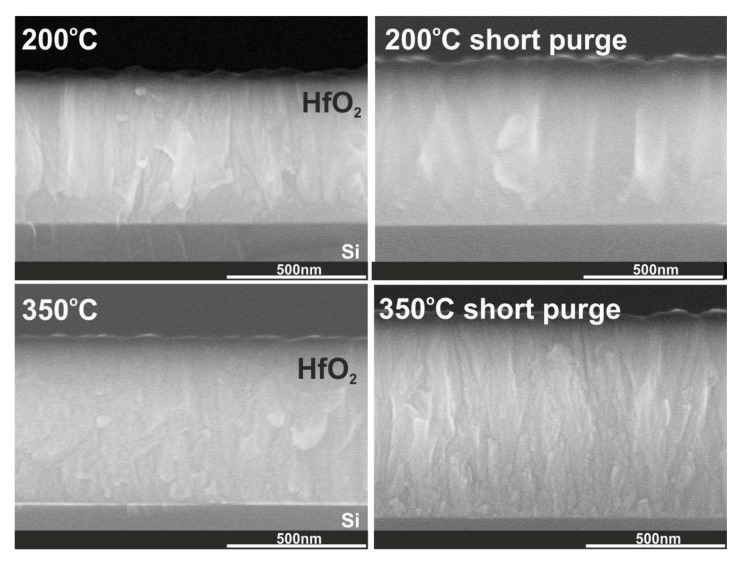
Cross-section SEM images for HfO_2_ thin films with thicknesses above 500 nm deposited at 200 and 350 °C on a Si substrate.

**Figure 11 materials-16-04077-f011:**
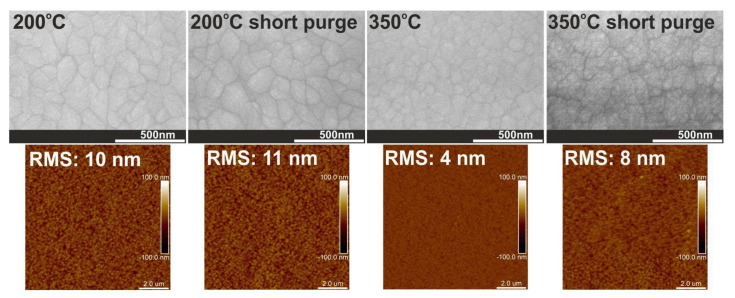
SEM images and AFM surface morphology with RMS values for HfO_2_ thin films with thicknesses of above 500 nm deposited at 200 and 350 °C on a Si substrate.

**Figure 12 materials-16-04077-f012:**
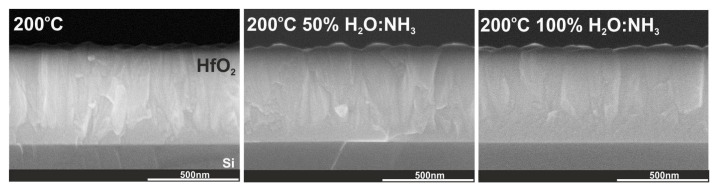
Cross-section SEM images for HfO_2_ thin films with thicknesses above 500 nm deposited at 200 °C on a Si substrate.

**Figure 13 materials-16-04077-f013:**
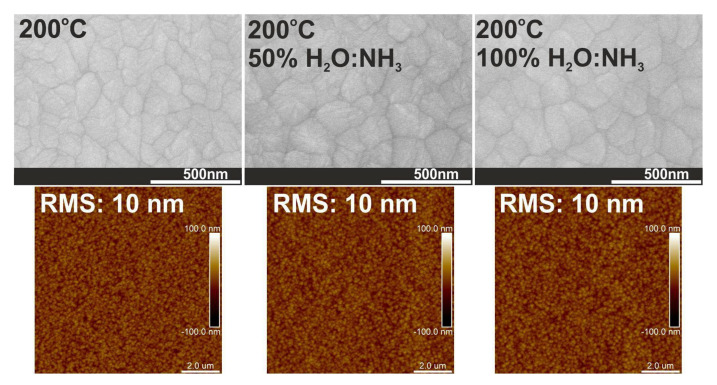
SEM images and AFM surface morphology with RMS values for HfO_2_ thin films with thicknesses above 500 nm deposited at 200 °C on a Si substrate.

**Figure 14 materials-16-04077-f014:**
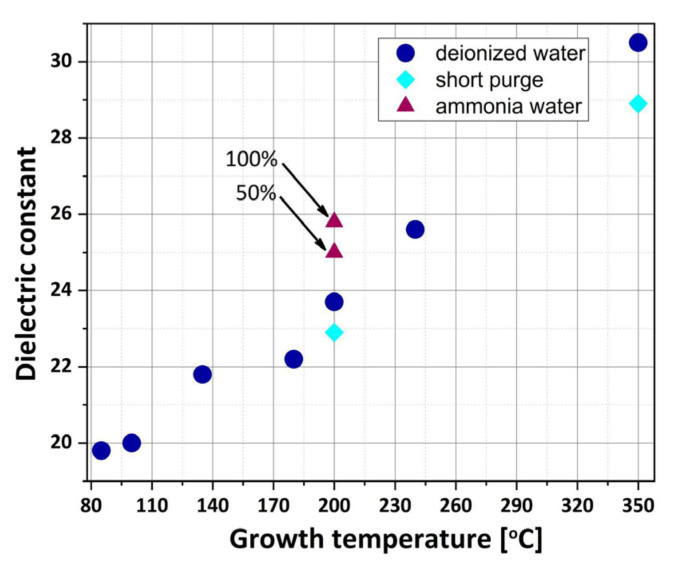
Dielectric constant values of HfO_2_ layers with thicknesses above 500 nm as a function of growth temperature.

**Figure 15 materials-16-04077-f015:**
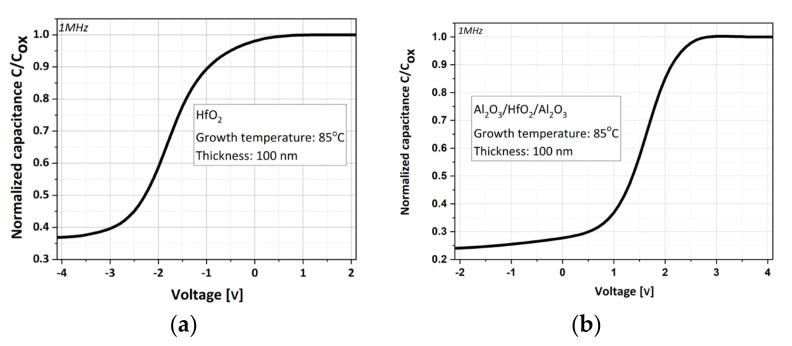
Normalized C–V characteristic for the 100 nm thick (**a**) HfO_2_ (**b**) Al_2_O_3_/HfO_2_/Al_2_O_3_ (5/90/5 nm) dielectric layer grown by ALD at 85 °C in the MOS structure.

**Table 1 materials-16-04077-t001:** EDX composition parameters of the above 500 nm-thick HfO_2_ films grown by using different growth conditions at different temperatures.

Growth Temperature [°C]	Changed Growth Condition *	O:Hf [atomic Ratio]	C [atom.%]	N [atom.%]
85	-	2.03	6.5	4.5
100	-	2.03	6	4.5
135	-	1.94	6	4.5
180	-	1.94	6	4.5
200	-	1.94	6	4.5
	short purge	1.91	5.8	4
	50% H_2_O:NH_3_	1.94	6	5
	100% H_2_O:NH_3_	1.94	6	5
240	-	1.94	6	4.5
350	-	2.00	6.5	4.8
	short purge	1.86	6	4.5

* The change was introduced in the growing conditions; details are described in Section 2.1.

**Table 2 materials-16-04077-t002:** Particle sizes on the surface and crystal sizes of above 500 nm-thick HfO_2_ films determined from AFM and XRD measurements, respectively.

Growth Temperature [°C]	Changed Growth Condition *	Particle Size on Surface [nm]	Crystals Size ± 4 34° (020) [nm]
85	-	2	-
100	-	83	29
135	-	65	38
180	-	43	38
200	-	33	39
	short purge	34	37
	50% H_2_O:NH_3_	38	37
	100% H_2_O:NH_3_	38	38
240	-	30	40
350	-	17	39
	short purge	27	39.5

* The change was introduced in the growing conditions; details are described in Section 2.1.

## Data Availability

Not applicable.

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
