# Peer review of "Atomic Layer Deposition of HfO2 Films Using TDMAH and Water or Ammonia Water"

_materials, 2023, doi:10.3390/ma16114077_

Round 1

Reviewer 1 Report

The manuscript is good and can be published in the current form  

The manuscript is good and can be published in its current form  

Author Response

Dear Reviewer, Thank You for this positive review.

Best regards.

Author Response

Dear Reviewer,

  1. In Fig. 4, the quality of elemental mapping images is unacceptable. It is too rough.

Thank You for this remark. The visibility of the scale was improved in Fig. 4. EDX obtained element distribution maps for only small orange square areas with nanometer precision.

  1. For better comparison, the standard XRD lines are suggested to be added at the bottom/top of the XRD figures.

Thank You for this remark. We improved XRD Figures according to Your suggestion.

  1. The surface chemistry of the resultant films should be investigated by XPS.

Thank You for this suggestion. In this paper, we have presented chemical analysis by an energy-dispersive X-ray spectrometer (EDX) in Table 1. We agree that You have great idea because XPS measurements provide a lot of interesting information, but this is an analysis for another publication.

  1. The authors claim that the increased presence of nitrogen in hafnium layers caused the increase in particle size on the HfO2 surface from 33 to 38 nm and the decrease in crystal size in films from 39 to 37 nm. The inhibiting crystallization and defects related to nitrogen in these oxide films contribute to the above argument. The related XPS analysis should be provided to support their statements.

Thank You for this comment. You pointed out HfO2 layers grown at 200 °C with replacing deionized water (H2O) with ammonia water (H2O:NH3) in ALD processes. Particle sizes on surfaces of HfO2 films were determined by AFM, whereas crystal sizes of HfO2 films were calculated using the XRD data. Therefore, the increase in particle size on the HfO2 surface and the decrease in crystal size in films indicate that the use of H2O:NH3 as an oxygen precursor reduced structural order in films. We suppose that these changes are most probably due to inhibited crystallization and defects related to nitrogen by introduction of the – NHx groups from ammonia water to these oxide films. It can be assumed that, as result of ALD reactions with H2O:NH3 as oxygen source, the – NH group is introduced exactly at the oxygen site of the HfO2 structure, as similarly presented in our previous work [Guziewicz, E., Przezdziecka, E., Snigurenko, D., Jarosz, D., Witkowski, B. S., Dluzewski, P., & Paszkowicz, W. (2017). Abundant acceptor emission from nitrogen-doped ZnO films prepared by atomic layer deposition under oxygen-rich conditions. ACS Applied Materials & Interfaces, 9(31), 26143-26150.]. We do not have the ability to perform XPS measurements of these samples in such a short time, but the presented EDX measurement results. For better understanding, we have made changes to the text. 

Kind Regards

Reviewer 3 Report

The authors present a paper on ALD deposition of HfO2 with the precursor TDMAH and water, in which they characterize the electrical and crystallographic properties of the deposited materials as a function of the process temperature. The background of the paper and the presented results do not seem to me very original, there is already a lot of literature on the subject, and the results obtained here confirm everything that has already been published.

However, the originality here may come from the thickness of the deposited films (> 500 nm) which seems to me to be huge considering the process for which usually the HfO2 layers are well below 100 nm!

So, I propose to publish the paper, I just have some remarks:

- can you justify the interest of such thick layers (application). Can you also give in your paper the time of the process (according to my calculations almost one day)

- page 7 figure 2, you use the term growth rate, I prefer the term growth per cycle which is the one used by the community and seems more logical with what you draw

- By changing the temperature from 85 to 240C the thickness goes from 650 to 520 nm (ratio 1.25) while the ratio according to your GPC should be 1.17 (1.4/1.2). Can you explain the difference? Do you have the same number of cycles?

- you mention a breakdown field for HfO2 of only 1MV/cm when this material is more likely to have a breakdown field around 5 MV/cm. Can you comment on your reliable value?

Author Response

Dear Reviewer

1. can you justify the interest of such thick layers (application). Can you also give in your paper the time of the process (according to my calculations almost one day)

Thank You for these remarks. Optical applications such as plasma displays, light emitting diode emitters, solid state lasers, and various scintillators need thick crystal layers. These applications are often based on wide-gap materials, such as HfO2 thin films, with a significant thickness of 200-700 nm and often activated with rare earth ions. Information and references about it have been added to the text.

The time of the process have been added to the text. Yes, the duration of a complete ALD process was about 4-24 hours for the 100–740 nm thick HfO2 films.

2. page 7 figure 2, you use the term growth rate, I prefer the term growth per cycle which is the one used by the community and seems more logical with what you draw

Thank You for this remark. We changed this term according to Your suggestion.

3. By changing the temperature from 85 to 240C the thickness goes from 650 to 520 nm (ratio 1.25) while the ratio according to your GPC should be 1.17 (1.4/1.2). Can you explain the difference? Do you have the same number of cycles?

Thank You for these remarks. Yes, we used the same number of cycles. The ratio difference is due to the rounding of decimal places in the GPC calculation. For example: for a film at 85 oC with 650 nm

GPC = 1.444444444444444 = 1.4(4)= ~1.4

4. you mention a breakdown field for HfO2 of only 1MV/cm when this material is more likely to have a breakdown field around 5 MV/cm. Can you comment on your reliable value?

Thank You for this remark. The obtained electrical parameters are dependent on many factors of MOS structures. The results of dielectric strengths for MOS structures with areas of 0.04–0.09 cm2 and with 100 nm thick HfO2 and composite Al2O3/HfO2/Al2O3 (5/90/5 nm) layers as gate dielectrics on n-Si substrates (10−2 Ωcm resistivity) were presented in this work. The dielectric strength of HfO2 layer was 1 MV/cm, while that of a layer of HfO2 with Al2O3 was 4 MV/cm. For comparison, dielectric strength was even higher than 6 MV/cm for MOS structures with HfO2 layers but with areas of 0.00049 cm2, as shown in our previous work [Gieraltowska, S., Wachnicki, L., Witkowski, B. S., Mroczynski, R., Dluzewski, P., & Godlewski, M. (2015). Characterization of dielectric layers grown at low temperature by atomic layer deposition. Thin Solid Films, 577, 97-102.]

Kind Regards

Round 2

Reviewer 2 Report

The revision can be accepted for publication.

Author Response

Dear Reviewer, Thank You for positive revision of our manuscript.